# Adaboost Algorithm in Artificial Intelligence for Optimizing the IRI Prediction Accuracy of Asphalt Concrete Pavement

**DOI:** 10.3390/s21175682

**Published:** 2021-08-24

**Authors:** Changbai Wang, Shuzhan Xu, Junxin Yang

**Affiliations:** School of Civil Engineering and Architecture, Anhui University of Science and Technology, Huainan 232001, China; shuzhanww@163.com (S.X.); 15856995257@163.com (J.Y.)

**Keywords:** MEPDG, LTPP, IRI prediction, decision tree, AdaBoost regression

## Abstract

The international roughness index (IRI) for roads is a crucial pavement design criterion in the Mechanistic-Empirical Pavement Design Guide (MEPDG). However, studies have shown that the IRI transfer function in the MEPDG is simply a linear combination of road parameters, so it cannot provide accurate predictions. To solve this issue, this research developed an AdaBoost regression (ABR) model to improve the prediction ability of IRI and compared it with the linear regression (LR) in MEPDG. The development of the ABR model is based on the Python programming language, using the 4265 records from the Long-Term Pavement Performance (LTPP) that include the pavement thickness, service age, average annual daily truck traffic (AADTT), gator cracks, etc., which are reliable data that are preserved after years of monitoring. The results reveal that the ABR model is significantly better than the LR because the correlation coefficient (*R*^2^) between the measured and predicted values in the testing set increased from 0.5118 to 0.9751, and the mean square error (MSE) decreased from 0.0245 to 0.0088. By analyzing the importance of variables, there are many additional crucial factors, such as raveling and bleeding, that affect IRI, which leads to the weak performance of the LR model.

## 1. Introduction

During the construction of asphalt concrete (AC) roads, the longitudinal profile of the pavement often changes due to the influence of the construction conditions. Moreover, in the long-term driving process, the outline shape will continue to change under heavy vehicle loads and harsh environments, visually manifesting as the pavement becomes rough. Vehicles running on uneven roads not only affect the safety and comfort of the passengers or drivers, but also increase operating costs (such as increasing fuel consumption, reducing driving speed, and extending the travel time) [1] and at the same time, will accelerate the destruction of the pavement structure, affecting the service life and maintenance cycle of the pavement [2]. Therefore, with the continuous improvement of highway service quality requirements and the establishment of pavement management systems, roughness has become one of the most critical indicators of current pavement performance. Accurately assessing the condition of a gradually distressed road surface will be a vital task of the Department of Transportation (DOT) to ensure the safe and durable operation of the national road traffic network [3,4].

To better express and quantify the roughness of the pavement surface, researchers have introduced a one-dimensional pavement longitudinal profile index called the international roughness index (IRI), which is widely used in the research on road and traffic engineering [5,6,7,8,9]. In the Mechanistic-Empirical Pavement Design Guide (MEPDG), IRI is also used as a robust design criterion that is critical to measuring the performance of AC pavement. Whether it is accurately estimated or not directly affects the reliability of pavement design. However, from the actual effect, the performance of the MEPDG model is not optimistic because its transfer function cannot predict the IRI of AC pavement well. To solve this problem, researchers have been using data collected from sites in various states to build new models or to optimize transfer functions for improving the accuracy of IRI prediction, which is called local calibration [10,11,12,13,14]. 

The IRI transfer function in MEPDG combines road distress with pavement materials, structure, and traffic conditions linearly through field measured data to predict the IRI over time for AC pavement. However, factors affecting IRI are widespread, especially nonlinear factors, so the linear model is too simple to provide a convincing prediction result. Moreover, Thompson, Barenberg et al. pointed out that the statistics and practical structural model constructed in this way cannot explain the change of IRI well, and that further feeding into the transfer function will not produce an accurate prediction [15]. To improve prediction performance, we need to establish a new model that can cover various influencing factors (including nonlinearity) and build the relationship between IRI and multiple parameters.

On this topic, relevant researchers worldwide have conducted many studies and analyses, so this is not a new concept that being presented here [16,17,18,19,20,21]. The artificial intelligence (AI) algorithm can introduce nonlinear calculations, which are suitable for establishing AC road distress prediction models. For example, In 2003, Lin et al. found a three-layer ANN model by using the collected data for deep learning, in which the number of neurons of the input layer, hidden layer, and output layer was 14, 6, and 1, respectively, and analyzed the correlation between variables [18]. To explore the superiority of the ANN model compared to a single linear or nonlinear regression model, Chandra et al. designed three models. The results of their work showed that the IRI prediction performance of the ANN model is significantly better than the other two models regardless of whether it is used on the training or testing set [22]. However, the ANN model may not always perform well, especially in the face of deeper ANN structures and huge input data, where overfitting (the model performed well on the training set, but poor on the testing set) is more likely to occur, which leads to the decrease of the model’s prediction ability [23,24]. 

Besides ANN, Freund and Schapire [25] proposed the AdaBoost model based on the boosting algorithm in 1997, which is one of the most widely used and researched algorithms. In particular, through 300 rounds of boosting tests Schapire [26] indicated that AdaBoost often did not find the overfitting phenomenon, apparently in direct contradiction of what was predicted by the Vapnik–Chervonenkis [27,28] (VC) theory, which ensures that the model has excellent and stable prediction performance. Thus, several recent works have proposed AdaBoost in order to compensate for pavement engineering distress. For example, In 2018, Wang, Yang, et al. used the AdaBoost algorithm to detect cracks on cement pavement and AC pavement, and the results were able to cover more than 83% of the crack length [29]. Hoang and Nguyen proposed an alternative method to automatically conduct periodic surveys of road conditions through image processing based on AdaBoost and achieved a high crack classification accuracy of about 90% [30]. Raveling is one of the most common forms of asphalt pavement distress on American highway pavements. Based on AdaBoost, after large-scale verification and refinement, the loose detection method developed by Tsai, Zhao, et al. had been successfully applied to the entire Georgia interstate highway system [31]. Compared to the application of AdaBoost in these aspects of road engineering, IRI predictions have higher similarities because the unevenness of AC pavement is caused by factors, including cracks, ruts, and raveling, etc., which contain some linear or nonlinear relationships. AdaBoost can explain the existence of these relationships through complex calculations, which provides the possibility of its application in IRI prediction modeling.

This study develops an AdaBoost-based IRI estimation model that utilizes the 4265 records from the Long-Term Pavement Performance (LTPP), including the pavement thickness, service age, average annual daily truck traffic (AADTT), gator crack, etc. Compared to the LR model from the MEPDG, the novelty of this model is that it comprehensively considers the input variables and introduces nonlinear factors. The results reveal that the AdaBoost regression (ABR) model has better fitting performance and has a lower error rate than the LR model on both the training and testing sets. Therefore, the ABR model can better predict IRI, and it is also easy to interpret. By analyzing the importance of variables, there are many additional crucial factors, such as raveling and bleeding, that affect IRI, leading to the LR model’s weak performance.

## 2. IRI Transfer Function

### 2.1. Definitions

To better perform local calibration, that is, to adapt the IRI transfer function to the requirements of different regions, LTPP has been collecting research quality pavement performance data from in-service test sections across the U.S. and Canada. The calibration process is to choose different coefficients to obtain the equation that best represents the actual situation. From the point of view of mathematical relationships, it is to select appropriate coefficients to obtain specific IRI calculation equations. Equation (1) gives the transfer function calculated by the IRI in the MEPDG. When calibrating locally, we generally need to go through two processes. The first step is to determine the value of *C*_1,2,3,4_, and the second step is linearly combining these coefficients with the road parameters by substituting them into the transfer function.
(1)$$\begin{array}{l} IRI=IRI_{o}+C_{1}(RD)+C_{2}\left(FC_{\mathrm{Total}}\right)+C_{3}(TC)+C_{4}(SF)\\ SF=Age^{1.5}\left\{\ln\left[(Precip+1)(FI+1)p_{02}\right]\right\}+\left\{\ln\left[(Precip+1)(PI+1)p_{200}\right]\right\} \end{array}$$
where *IRI*_0_ is the initial IRI after construction, *SF* is the site factor, *FC*_Total_ is the area of fatigue cracking, *TC* is the length of transverse cracking, *RD* is the average rut depth, *C*_1,2,3,4_ are the calibration factors, *Age* is the pavement age, *PI* is the percent plasticity index of the soli, *FI* is the average annual freezing index, *Precip* is the average annual precipitation or rainfall, *p*_02_ is the percent passing the 0.02 mm sieve, and *p*_200_ is the percent passing the 0.0075 mm sieve.

### 2.2. IRI Local Calibration Efforts

Table 1 shows some fitting parameters of the IRI calibration results, where *R*^2^ and MSE represent the correlation coefficient and the mean square error, respectively. *R*^2^ is also called the coefficient of determination, which defines how many future samples may be expected by the regression model. Moreover, it also means that the variance ratio of the predicted response variable in the variable can be explored [32]. Its value ranges from 0 to 1.0, where 1.0 is the best. MSE is a measure that reflects the degree of difference between the estimated and the actual value. Equation (2) gives the specific calculation method of *R*^2^ and MSE. From the records listed in Table 1, it can be seen that the results of these calibrations are not satisfactory. For regions such as New Mexico, Arkansas, Kansas, and Ontario (Canada), the *R*^2^ is below 0.5, which means that these models cannot explain more than half of the sample data. Specifically in New Mexico, the MSE is as high as 0.12. Even if the *R*^2^ in some regions exceeds 0.5, such as in Arizona and Iowa, these indicators are only based on training data. In practice, it cannot behave like this in the face of unknown samples.
(2)$$\begin{array}{l} R^{2}=1-\frac{\sum {\left(y-\hat{y}\right)}^{2}}{\sum {\left(y-\bar{y}\right)}^{2}}\\ MSE=\frac{1}{n}\sum {\left(y-\hat{y}\right)}^{2} \end{array}$$
where *y* is the measured value, *ŷ* is the predicted value, and *n* is the total number of samples, *R*^2^ is the coefficient of determination, and *MSE* is the mean square error.

## 3. Overviews of the AdaBoost Method

### 3.1. Decision Tree

The decision tree is a practical machine learning classifier that requires supervised learning, which trains a given sample when the result is already known [40,41,42]. As shown in Figure 1, the decision tree mainly consists of the root node, decision node, and leaf node. The sample starts from the root node and is classified according to the rules of each layer until it can no longer be divided. When using a decision tree, you first need to construct the root node, which contains all of the training data. Second, an optimal feature needs to be selected, and the training data need to be divided into subsets according to this feature so that each subset can achieve the best classification under the current conditions. Next, the leaf nodes need to be constructed for the subsets that have been correctly classified, and these lead nodes needs to be assigned to the corresponding leaf nodes. If there are subsets that cannot be correctly classified, then new optional features need to be selected for them. These need to be continued to be divided, and corresponding tree nodes need to continue to be built until all of the training subsets are correctly classified or until there are not suitable features. Finally, each subset needs to be assigned to a lead node and needs to have a clear class. Commonly used decision trees are ID3 [43,44], C4.5 [45,46] and the classification and regression tree (CART) [47,48,49,50]. ID3 calculates the information gain of all of the possible features and selects the maximum value as the feature of the node. Then, leaf nodes need to be constructed from the different features in order to recursively generate decision trees for the lead nodes until there are no features left to be chosen. However, such mechanical recursive calculations will not continue until the tree generated is often very accurate in the classification of the training data but not as accurate on the test set. That is, overfitting will occur. C4.5 inherits the advantages of ID3 and has been improved. Pruning (cutting some subtrees or leaf nodes from the tree that has been generated and uses the root node or parent node as the new leaf node to simplify the classification tree model) is conducted during the tree construction process, and the classification rules that are generated are easy to understand and have a high accuracy rate. However, in constructing the tree, the data set needs to be scanned and sorted multiple times, which leads to the inefficiency of the algorithm. Regardless of ID3 or C4.5, the main solution is the classification problem, which is the processing of discrete values. CART has made significant improvements based on the C4.5 model and can handle continuous values in regression problems. In the pruning process, the generated tree is pruned with the verification data set, and the optimal subtree is selected, thereby effectively reducing the occurrence of overfitting. Moreover, CART is suitable for large-scale data sets, especially considering that the more complex the sample and the more variables that there are, the more significant its superiority is. Since CART can solve two types of problems: classification and regression problems, it is used as a weak classifier for IRI prediction. 

### 3.2. AdaBoost

A single decision tree is called a weak learner because of its limited capabilities. Researchers imagine whether a strong learner can be obtained if multiple weak learners are combined. Schapire [51] proved this conjecture in 1990 and laid the foundation for the boosting algorithm, which combines multiple weak learners in series. As shown in Equation (3), each time it adds a new tree model to the whole, the general tree will be eliminated, and only the strongest tree will be added. In this way, with the accumulation of iterative calculations, the overall model performance will gradually improve. However, there is a problem here. After the first basic tree model is obtained, some samples on the data set are correctly classified, but some are wrong. The AdaBoost algorithm is a simple weak classification algorithm improvement process, which improves data classification ability through continuous training. The first weak classifier is obtained by learning the training samples, and the wrong samples are combined with the untrained data to form a new training sample. Furthermore, the second weak classifier is obtained by learning this sample. The wrong sample is combined with the untrained data to form another new training sample, which can be trained to obtain the third weak classifier. After repeating this process many times, we can finally obtain the improved robust classifier. To increase the number of correct classifications, the AdaBoost algorithm gives different weights to the samples [52]. The correctly classified samples are provided relatively low weights, and the wrong ones must be increased, which forces the model to pay more attention to the misclassified samples [53]. Figure 2 describes the overall calculation process of the AdaBoost algorithm. When training each basic tree model, the weight distribution of each sample in the data set needs to be adjusted. Since each training data will change, the training results will also be different, and finally, all of the results are summed [26].

(3)$$F_{n}\left(x\right)=F_{m-1}\left(x\right)+argmin_{h}\underset{i=1}{\sum^{n}}\;L\left(y_{i},F_{m-1}\left(x_{i}\right)+h\left(x_{i}\right)\right)$$
where *F_n_*(*x*) is the overall model, *F_n-_*_1_(*x*) is the overall obtained in the previous round, *y_i_* is the prediction result of the *i*-th tree, and *h*(*x_i_*) is the newly added tree.

### 3.3. Framework of the ABR Model

#### 3.3.1. The Process of Building ABR Model

This section explains the overall establishment of the ABR model for estimating road IRI by using sample data including 21 (20 features and 1 target) columns of data such as the initial IRI (IRI_0_), time of service, the total thickness of the pavement, AADTT, etc. The data used for training is the most original, and none of them have been modified by an algorithm in order to prevent damage to the model’s practicality. As illustrated in Figure 3, the ABR framework consists of four parts: (1) randomly splitting training instances into training and testing subsets, (2) training an AdaBoost model on the training subsets, (3) making predictions on the testing subset, and (4) comparing the testing predictions to the testing targets to assess accuracy.

In the first step, the processed data are randomly divided into two data sets, employing one of them as the training set (80%) for model building. The other part is used as the testing set (20%) for model evaluation. The second step is to use the data in the training set for model training, during which cross-validation methods will be used for validation. In the third step, the build model is used to predict the samples on the testing set. In the last step, the difference between the the predicted value and the actual value to judge the performance of the model.

#### 3.3.2. Real AdaBoost or Gentle AdaBoost

To solve the IRI prediction regression problem, as mentioned above, CART is selected as the weak learner. On this basis, the commonly used AdaBoost models are the Real and Gentle algorithms. Real AdaBoost [54,55] uses a logarithmic function to map to the real number domain after each weak classifier outputs the probability that the sample belongs to a specific class, and Gentle AdaBoost [56] performs a weighted regression based on the least-squares in each iteration [57]. The difference between the two algorithms is that Real is mainly used for classification problems, while Gentle is good at solving regression problems. IRI prediction is a regression problem, so Gentle is more suitable for modeling. To verify this judgment, we attempted to compare the MSE of the two models by establishing 2000 estimators (trees), and the results are shown in Figure 4. Regardless of whether it is used on the training set or the testing set, Gentle has a lower error rate, and the data fluctuation range is small. After about 200 calculation interations, it can quickly converge, and the error curve tends to be stable.

#### 3.3.3. Loss Function

In the model, the predicted value of the regression equation and the actual value of the sample points are not one-to-one. There will be an error between each actual and predicted value, which is usually called an error term. We hope that the difference between the predicted formula and the actual value is as small as possible, so we define a way to measure the quality of the model, that is, the loss function (used to express the degree of difference between the prediction and the actual data). The loss function quantifies this error, and whether its selection is appropriate can affect subsequent optimization work. In the field of machine learning, common loss functions are linear, square, and exponential. We tried to establish 50 estimators to determine the appropriate loss function, and the results are shown in Figure 5, wherein the exponential score is better than the other two functions. After about 20 iterations in the training set, the curve tends to be stable.

## 4. Data Preparation 

### 4.1. Data Acquisition from LTPP

Collecting comprehensive and accurately labeled data is necessary for model training. Hence, the data used comes from LTPP, which gathers research quality pavement performance data from in-service test sections across the United States and Canada. This study initially obtained more than 11,000 IRI records from 62 states in the United States and Canada. However, due to the lack of average annual daily truck traffic (AADTT) data, only 4265 samples with AADTT data were used in the end. Figure 6 depicts the source distribution of these data.

### 4.2. Predictor Variables Selection

The collected samples contain many variables, but not every variable can improve the model’s performance, and unrelated variables reduce the accuracy and computational efficiency of the model. Equation (4) gives the calculation method for the variable importance. First, the accuracy of the out of bag (OOB) data set that is part of the data set is calculated, and the input variables are randomly arranged, and the OOB accuracy of the decision tree is recalculated to obtain the difference between the two arrangements [58]. To remove irrelevant variables, we selected relative importance to evaluate the importance of variables and eliminated variables with values less than zero. The steps are as follows: (1) calculate the importance values of all variables and sort them in descending order of relative importance values, (2) divide the variables evenly into n groups and keep the last set of variable rankings and values, (3) calculate the remaining variable importance values and sort them in descending order of relative importance value, (4) repeat step (3) until the calculation of these groups of variables is completed, and (5) repeat the simulation 100 times and take the mean value of relative importance of these 100 times as the variable importance value.

(4)$$VI=\frac{1}{N}\sum_{k}\;\left(errOOB_{k}^{l}-errOOB\right)$$
where N is the number of trees, ${errOOB}_{k}^{l}$ is the error of predictorlon the permuted sample, and *errOOB* is the error of a single tree in the OOB sample.

The above process results show that the variable importance values of average wind velocity, average temperature, average cloud cover, and shortwave surface average are all less than zero, so they are all eliminated. Moreover, to further optimize the input variables, we introduced stepwise regression analysis to filter the variables. The basic idea of stepwise regression is to introduce variables, test them one by one, and eliminate variables that are no longer significant due to new variables. If neither significant variable is selected into the equation, and if all insignificant independent variables are excluded from the regression equation, the process ends. The specific process is shown in Figure 7. We can see that after the execution of this process, by comparing the OOB values of the variables before and after, the unimportant variables can be eliminated, and the minimum OOB can be obtained. After testing, we successfully eliminated the pumping length, pavement width, air voids, resilient modulus, aggregate percentage, and binder content of the four variables. Therefore, the model ultimately leaves 20 relevant variables. Table 2 describes the specific content of these variables in terms of structure, performance, traffic, and climate.

### 4.3. Data Set Allocation

However, no matter how good the model is in all aspects of the training process, our ultimate goal is to achieve accurate prediction results in practice. Usually, the error that we produce when the model is applied to reality is called the generalization error, so we need to reduce the generalization error to improve the environmental adaptability of the model. Nevertheless, it is not realistic to understand the model’s generalization ability directly by using the generalization error as a signal. This requires frequent interaction between the model and the reality, which increases the difficulty and cost of modeling. A better way is to split the data into two parts: the training set and the testing set. The data in the training set are used for model training. Then, the error of the trained model on the testing set is calculated as an approximation of the generalization error, so we only need to reduce the model’s error on the testing set when optimizing the model. In this study, we randomly divided the data into two parts, where 3412 (80%) samples are used for model training, and the remaining 853 (20%) are used for testing.

## 5. ABR Model Construction

The transfer function given in MEPDG considers 10 factors as input variables when predicting IRI. On the contrary, this ABR model also considers other essential factors for comprehensive construction, listed in Table 1.

As mentioned above, this model training uses 4265 IRI samples from LTPP, and each sample contains 20 variables and 1 measured value of IRI. To achieve better prediction results, selecting appropriate hyperparameters, including CART parameters and ABR frame parameters, is necessary. The parameters that CART needs to determine mainly include the Max depth, Min samples split, and Min samples leaf nodes. The frame parameters of ABR include base estimator, loss, N estimators, and learning rate. Considering the limited number of samples, it would be very wasteful to set the verification set separately. To solve this problem, we used four-fold cross-validation, as shown in Figure 8. When verifying a certain result, the whole process needs to be divided into four steps. In the first step, use the first three iterations as the training set and the last interation as the validation set to achieve a result. By analogy, each time uses a different three from the training set, and the rest are used as the validation set. After completing the four steps using this method, four results are obtained, each of which corresponds to each small part, and the combination contains all of the data in the original training set. The final four results are then averaged to obtain the final model evaluation result. Cross-validation can better evaluate the model and make the results more accurate.

In theory, to find a suitable combination of hyperparameters, all possible values should be listed in turn, but this will consume a lot of time for the traversal and reduce efficiency. In response to this problem, a common strategy is a random search, which randomly tests all of the possible values when optimizing hyperparameters to find a position that is roughly close enough to the optimal solution. Although its results are not as accurate as a comprehensive search, it dramatically improves the iterative efficiency, especially when facing large-scale data sets. When considering the hyperparameters of CART, we limit Max depth to between 4 and 16 and delineate the range of Min samples split from 2 to 10 because Max depth and Min samples split that are will also reduce modeling efficiency and are not conducive to learning the characteristics of the sample. For the choice of Min samples leaf, we considered a larger value in the early stage of modeling, but it caused a severe overfitting problem, so it was finally determined to be between 2 and 5. Moreover, a too large learning rate (step size) will cause the model to miss the minimum value when calculating the loss, so the learning rate of this model is set to be low (0.001). After defining the range of these hyperparameters, the random search method (in the Python language, using the RandomizedSearchCV function under the random forest class) is then used to determine the specific value. Based on the fast convergence characteristics of the AdaBoost algorithm, the model’s error stabilizes after 80 iterations. Table 3 lists the names and specific values of these parameters.

## 6. Model Result and Analysis

### 6.1. Model Performance Evaluation

After going through the building steps of the above model, we need to evaluate the model’s quality. This time, the selected metrics are the *R*^2^ and MSE. To better evaluate the ABR model, the training results will be compared with the LR model in the MEPDG, and the results are shown in Figure 9. The LR had an *R*^2^ of 0.5588 and a MSE of 0.087 in the training set, versus 0.5118 and 0.0249, respectively, in the testing set. As the *R*^2^ and MSE shown, this simple ABR model achieved a much better predictive performance than LR. Compared to the LR, the training *R*^2^ of the ABR is improved by more than 78%, while the MSE decreased by 98%. In contrast to the LR, the testing *R*^2^ of the ABR is also improved by 90%, while its MSE is reduced by 65%. Table 4 lists the more detailed results of the two models. The results demonstrate that a simple ABR can be more predictive and capable of handling the IRI than a LR model. 

### 6.2. Model Interpretation

Numerical analysis results show that ABR can accurately predict the IRI of asphalt concrete pavement, and it performs well in the test set so that the model performs well in preventing overfitting. However, the nature of it cannot be revealed well from the numerical results. We need to implement the actual interpretation of this model. To this end, we use the importance analysis method in the random forest module, which can help us understand the relationship between the input variables and the actual IRI to assume part of the interpretation of the model [59,60]. It can be seen from the descending order graph of variable importance in Figure 10 that IRI_0_ is the most important factor affecting IRI, which is reflected in the IRI transfer function of MEPDG. The influence of the transverse cracks and AADTT is also very significant because the formation of transverse cracks will increase the vertical vibration of the driving vehicle, and the number of repeated actions of the vehicle will cause the permanent deformation of the road surface. Moreover, the traffic (ESAL), temperature (freeze index), and service age were also highly correlated with the IRI. Of course, some variables are easily overlooked, such as patches, which will cause a drop on the pavement to become uneven.

Further, Figure 11 draws a matrix scatter plot of the correlation between variables. Because there are too many variables to display, we have selected the most representative variables from each category for analysis. It can be seen that IRI_0_ has a solid linear correlation with IRI, which is consistent with the conclusions we have previously obtained. In addition, other variables do not obey such an apparent linear relationship, indicating that nonlinear factors are widespread. This is also the key to the difference between ABR and MLR and why ABR is highly predictive.

## 7. Conclusions

In this paper, the authors have examined and analyzed an ABR model for IRI estimation in AC pavements, aiming to optimize its configuration and some influencing factors for maximizing the resulting quality and the estimation reliability. Further, several aspects of adjusting the accuracy of the ABR have been considered, including weak learners and loss functions, etc., to arrive at recommendations for proper model optimization. The considered model data comes from the LTPP large-scale pavement information database and includes some parameters that affect the performance of the pavement structure, such as structure, climate, traffic, and performance variables. After the iterative calculation of 4265 samples from the LTPP and the analysis of the correlation between the variables, the following conclusions can be drawn:The *R*^2^ and *MSE* of the ABR model on the testing set are 0.9571 and 0.0088, respectively. Hence, the ABR model has high accuracy and predictability, so the model’s overfitting is well controlled;Compared to the LR, the testing *R*^2^ of the ABR is improved by 90%, while its *MSE* is reduced by 65%. The results demonstrate that a simple ABR can be more predictive and capable than a LR model;By analyzing the importance of the variables to the model, we can see the degree of influence of each variable. IRI_0_ is the largest, followed by transverse and AADTT. Moreover, there are many additional crucial factors such as raveling and bleeding that affect IRI, leading to the LR model’s weak performance;One of the reasons for the low predictive ability of the LR model in MEPDG is that it does not consider nonlinear influencing factors, which can be well improved by an ABR model for future pavement design.

In addition, the numerical results prove that the IRI_0_ (initial IRI) is the most important influencing factor, which is consistent with the transfer function in MEPDG so that the ABR model can be explained by practice. The results and discussion are helpful in optimizing the IRI prediction model to evaluate the performance of the pavement design structure. It is recommended to conduct further research work, hoping to collect more accurate road performance data through innovative high-performance testing equipment and expand the data set (in order to make the data input to the model better and more comprehensive) to further develop the potential of the model.

## Figures and Tables

**Figure 1 sensors-21-05682-f001:**
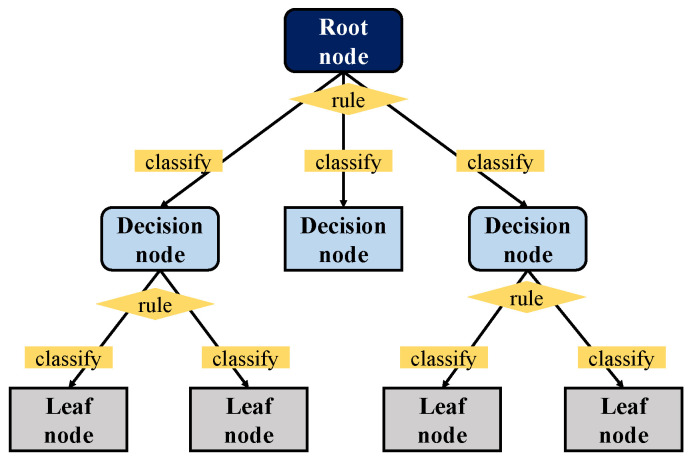
Decision tree.

**Figure 2 sensors-21-05682-f002:**
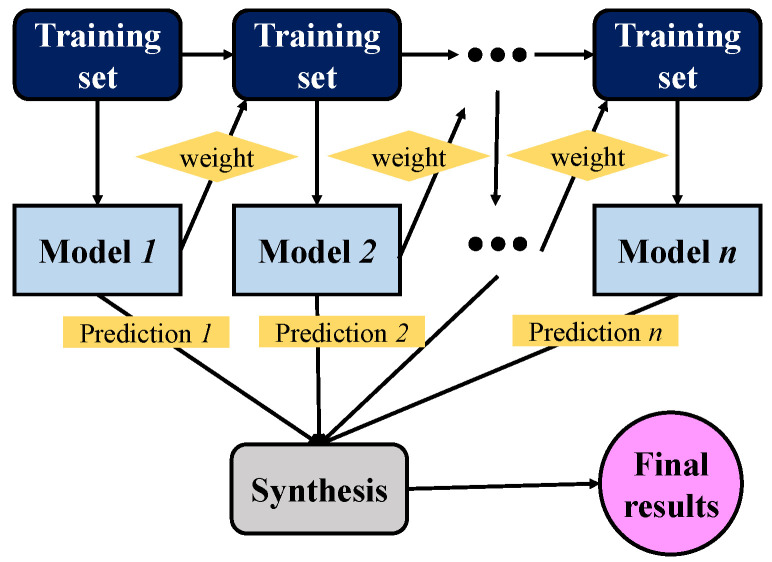
AdaBoost algorithm calculation process.

**Figure 3 sensors-21-05682-f003:**
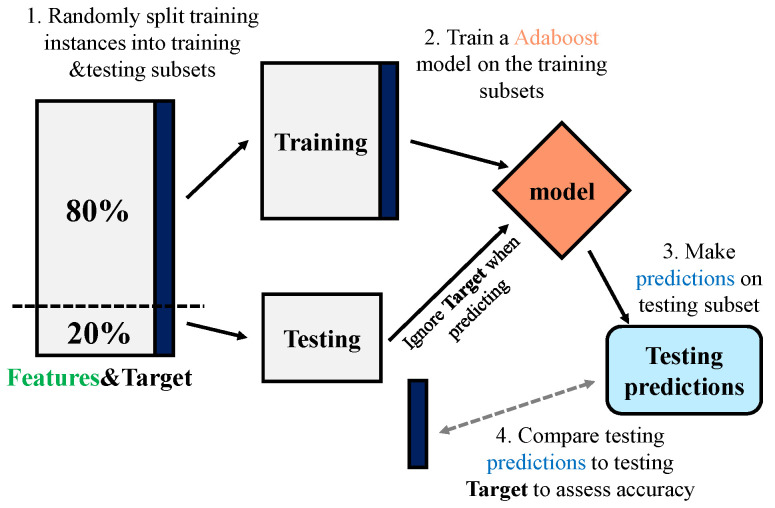
The AdaBoost model building process.

**Figure 4 sensors-21-05682-f004:**
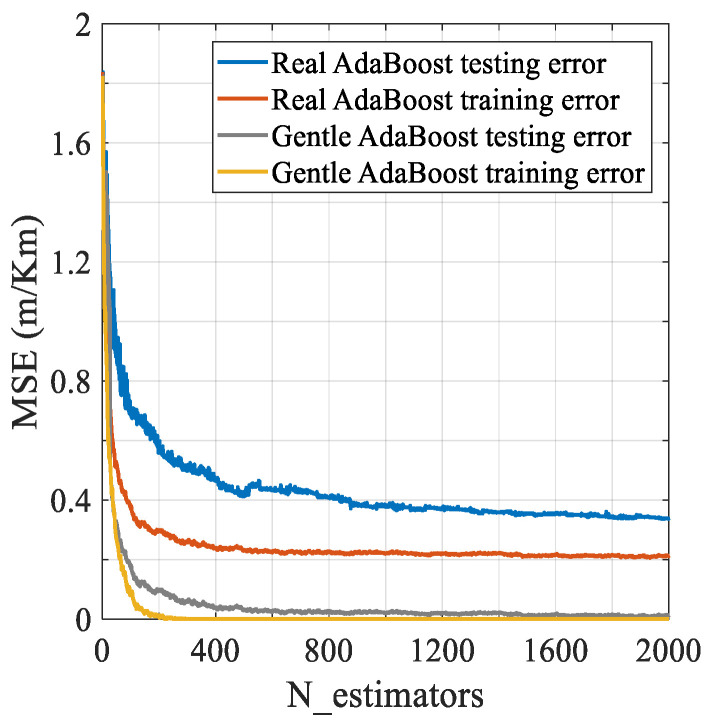
Real and Gentle error decay curve.

**Figure 5 sensors-21-05682-f005:**
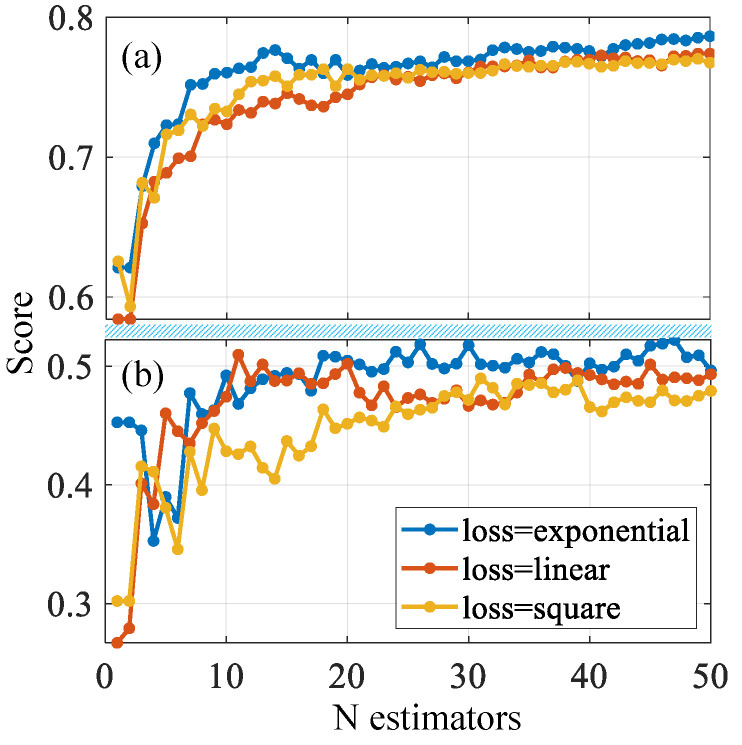
Different loss function score curves: (**a**) training set and (**b**) testing set.

**Figure 6 sensors-21-05682-f006:**
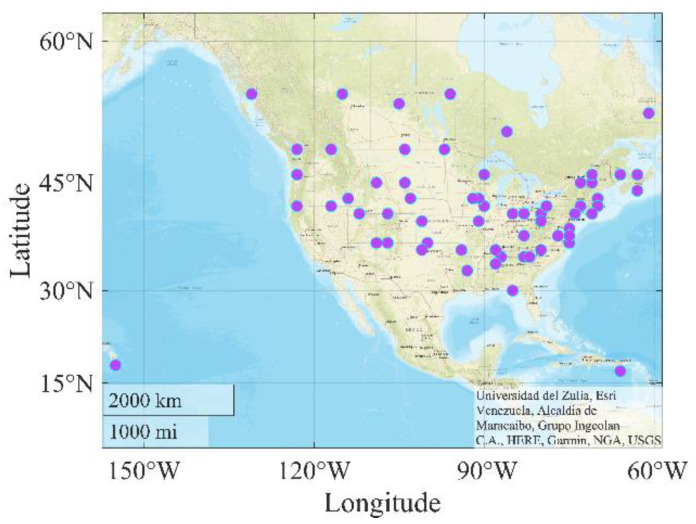
Distribution of data sources.

**Figure 7 sensors-21-05682-f007:**
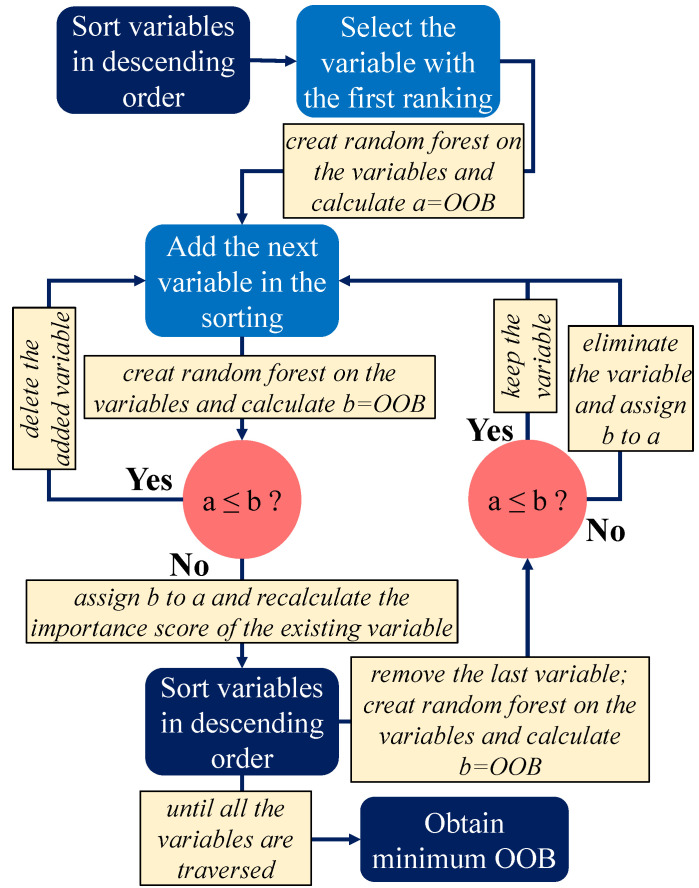
Stepwise regression.

**Figure 8 sensors-21-05682-f008:**
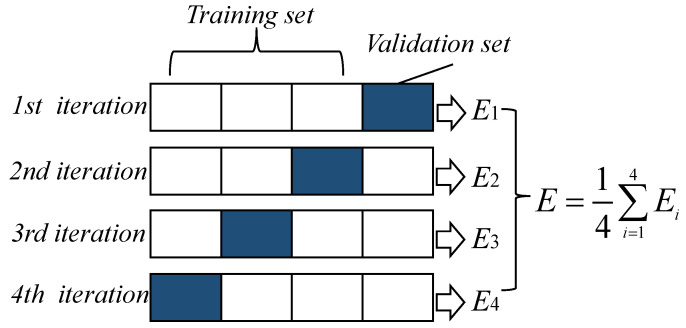
Cross-validation.

**Figure 9 sensors-21-05682-f009:**
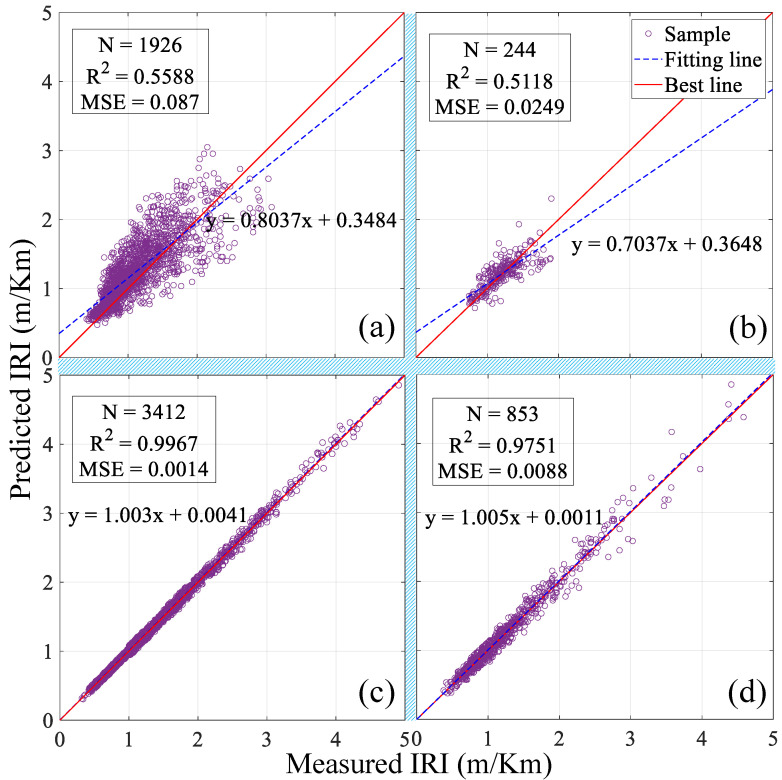
IRI predicted and measured values: (**a**) LR model on the training set, (**b**) LR model on the testing set, (**c**) ABR model on the training set, and (**d**) ABR model on the testing set.

**Figure 10 sensors-21-05682-f010:**
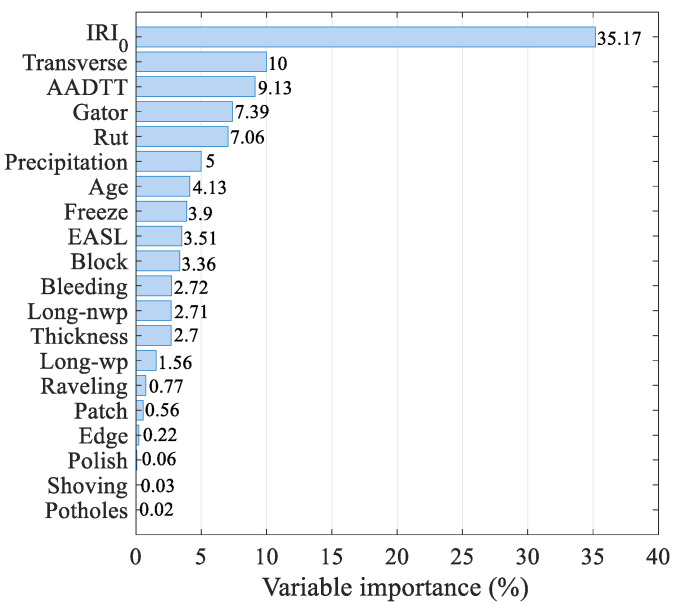
Importance of input variables.

**Figure 11 sensors-21-05682-f011:**
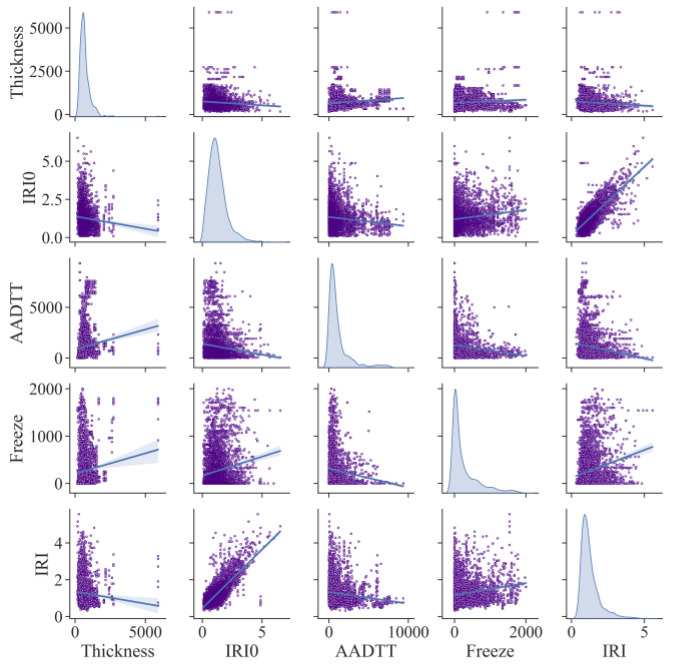
Variable correlation analysis matrix diagram.

**Table 1 sensors-21-05682-t001:** A summary of efforts in calibrating IRI prediction.

Literature	Region	*R* ^2^	MSE	Samples
AASHTO [33]	National (U.S.)	0.5118	0.0249	244
Schram, Abdelrahman [34]	National (U.S.)	0.621	0.0235	670
Tarefder, R.-R. [10]	New Mexico (U.S.)	0.326	0.12	85
Xiao and Wang [35]	Arkansas (U.S.)	0.476	0.063	193
Sufian [36]	Kansas (U.S.)	0.22	0.052	90
Souliman, M. [37]	Arizona (U.S.)	0.543	0.0355	178
Ceylan, K. [38]	Iowa (U.S.)	0.685	0.0196	65
Gulfam, Y. 13]	Ontario (Canada)	0.438	0.0775	90
Yuan, L. [39]	Ontario (Canada)	0.578	0.03	15

**Table 2 sensors-21-05682-t002:** Explanation of input variables.

Category	Variable	Explanation	Unit
Structure	Thickness	The total thickness of pavement	mm
Performance	Gator	Area of alligator (fatigue) cracking.	m^2^
	Block	Area of block cracking.	m^2^
	Edge	Length of low severity edge cracking.	m
	Long-wp	Length longitudinal cracking in wheel path.	m
	Long-nwp	Length non-wheel path longitudinal cracking.	m
	Transverse	Length of transverse cracking.	m
	Patch	Area of patching	m^2^
	Potholes	Area of potholes	m^2^
	Shoving	Area of shoving, localized longitudinal displacement of the pavement surface.	m^2^
	Bleeding	Presence of excess asphalt on the pavement surface which may create a shiny, glass-like reflective surface.	m^2^
	Polish	Area of polished aggregate (binder worn away to expose coarse aggregate).	m^2^
	Raveling	Wearing away of the pavement surface caused by the dislodging of aggregate particles and loss of asphalt binder.	m^2^
	Rut	Depth of rut	mm
	IRI_0_	The first IRI when the road is put into use	m/Km
	Age	Road service time	\
Traffic	AADTT	Average annual daily truck traffic	\
	ESAL	The annual average of equivalent single axle load in the LTPP lane.	\
Climatic	Freeze	Annual average freeze index	\
	Precipitation	Annual average precipitation	mm

**Table 3 sensors-21-05682-t003:** Selection of model parameters.

	Parameter	Explanation	Value
CART	Max depth	maximum depth of decision tree	12
	Min samples split	minimum number of samples required for subdividing internal nodes	8
	Min samples leaf	minimum number of samples for leaf nodes	3
ABR frame	Base estimator	weak regression learner	CART
	Loss	loss function, there are three choices of linear, square and exponential	exponential
	N estimators	maximum number of iterations of the weak learner	80
	Learning rate	the step size of the update parameter, too small will slow down the iteration speed	0.001

**Table 4 sensors-21-05682-t004:** Performance comparison of LR and ABR.

Model		*R* ^2^	MSE	Fitting Equation
LR	Training set	0.5588	0.0870	y = 0.8037x + 0.3484
	Testing set	0.5118	0.0249	y = 0.7037x + 0.3648
ABR	Training set	0.9967	0.0014	y = 1.0030x + 0.0041
	Testing set	0.9751	0.0088	y = 1.0050x + 0.0011

## Data Availability

Data are contained within the article.

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
