# Peer review of "Adaboost Algorithm in Artificial Intelligence for Optimizing the IRI Prediction Accuracy of Asphalt Concrete Pavement"

_sensors, 2021, doi:10.3390/s21175682_

Round 1

Reviewer 1 Report

Thank you for your paper. It is fairly well written. It is noted that it builds on other research that has used the AdaBoost algorithm to address pavement management issues. Its focus, the Mechanistic-Empirical Pavement Design Guide transfer function with respect to road roughness prediction, with a  focus on improving International Roughness Index (IRI) prediction modelling.

The discussion in the paper should clearly show how it makes a clear advance on previous research in this field.

The discussion of the AdaBoost approach as applied to this research (Sections 3) can be further developed. Further discussion is required, for example, of the decision tree in Figure 1, including the relationships between the nodes. There also should be further discussion in Section 3,2 about how the Adaboost methodology is able to address the issues with the Classifications and Regression tree approach. The discussion with respect to the model building process (Section 3.3.1) is reasonable.

The paper could be enhanced with a  discussion why the Python programming language was used for developing the AdaBoost model.

The discussion about data preparation (Section 4) is fairly good. However, in Section 5 (ABR Model Construction), there sould be additional discussion about how the model is constructed and the four-fold cross validation process construction.

Should Table 1 in Line 234 be Table 2? 

The figures showing results (Section 6) are presented quite well.

A further step in this research would be to test the methodology to the prediction of IRI on pavements in areas other than the United States and Canada. This would provide  an additional measure of validation. The approach could also be tested on pavements designed using systems other than the MEPDG.

 Finally, the list of references, although comprehensive, should include co-authors of papers as well as their first authors.

Reviewer 2 Report

The manuscript developed the AdaBoost regression to improve the prediction accuracy of the IRI of the asphalt concrete pavement. The presentation of results through figures and tables is logical, and the English language is correct. The topic of the paper is important and of potential interest to the readers of Sensors journal. However, the authors should clarify some points before the manuscript can be considered for publication.

  1. In addition to provided text in the current form of the introduction, the authors should answer the following two questions: (1) what algorithm is proposed in this paper, and (2) what are the main contributions.
  2. In Figure 5, the training scores and testing scores for different loss functions are suggested to divide into two figures.
  3. In Figure 7, the legends are too small to distinguish.
  4. The LR and ABR model have different training samples. Thus, it may not explain the feasibility of comparing results.
  5. There are many input variables in the ABE model. In practice, irrelevant variables degrade the accuracy and computational efficiency. As reported by previous literature (Journal of Machine learning, 2003, 3 (Mar): 1399-1414; Journal of Structural Engineering, 2021, 147(1):04020297), unnecessary predictor variables should be eliminated to obtain an efficient regression model. The random forest presented in the above references is recommended to select the important predictors.
  6. How to determine the hyperparameters of the proposed regression model?
  7. It is not sufficient to compare the linear regression. This paper should add the analysis of comparative results with the latest related algorithms.

Reviewer 3 Report

The paper presents a research dealing with the development of an AdaBoost regression model with the aim of improving the prediction ability of the IRI in comparison to the linear regression in MEPDG.  The topic is interesting in the field of pavement construction and management due to the positive implications on performance and costs of pavements. The topic is well posed and the model construction detailed described together with the results. To calibrate the model and assess its capability to predict the IRI, authors use data from LTPP referred to in-service test sections across United States and Canada. Is it possible to improve the model performance evaluation using a data sample referred to a road sections more homogeneous for location and function? Or can this compromise the model's ability to generalize? Please discuss this aspect.

Round 2

Reviewer 1 Report

Thank you for your paper.

  1. The authors should clearly distinguish the advances in this paper from previous research.

This topic is now addressed in lines 96 to 102. It does not appear to be further addressed in the conclusion.

  1. Several suggestions for improvement have been made. There should be additional discussion in Section 3.1 of the Classification and Regression Tree and its components, and how applying the AdaBoost algorithm to the calculation process improves the model learning process.

The authors have discussed the Classification and Regression Tree and its components in lines 146 to 174.

  1. The paper could be enhanced with a discussion of why the Python programming language was used for developing the AdaBoost model.

This request has not been addressed. It is not a significant issue.

  1. The discussion about data preparation (Section 4) is good. However, there should be further discussion about construction of the model (Section 5).

This topic has been addressed 293 to 319.

  1. The results (Section 6) are presented fairly well. Conclusions seem consistent with the evidence and address the research objectives. 

    No further requirement.

  2. A further step in this research would be to test the methodology to the prediction of IRI on pavements in areas other than the United States and Canada. This would provide an additional measure of validation. The approach could also be tested on pavements designed using systems other than the MEPDG.

While lines 391 to 398 discuss further development of the methodology, this specific request has not been addressed. It is recommended that the authors give further consideration to evaluating the methodology on additional pavement data sets. 

  1. The list of references, although comprehensive, should include co-authors of papers as well as their first authors.

Reviewer 2 Report

The authors addressed part of the reviewer's concerns. Although this paper determines the input variables using the random forest according to the reviewer's suggestion, the detailed procedure should be presented. Furthermore, the references recommended in previous comments are suggested to enhance the application of this method, e.g., Journal of Machine learning, 2003, 3 (Mar): 1399-1414; and ASCE  Journal of Structural Engineering, 2021, 147(1):04020297. 
